# Spatial Impressions Monitoring during COVID-19 Pandemic Using Machine Learning Techniques

**Talal H. Noor** [1,†], **Abdulqader Almars** [1,†], **Ibrahim Gad** [2,†], **El-Sayed Atlam** [1,2,†] and **Mahmoud Elmezain** [1,2,*,†]

1   College of Computer Science and Engineering, Taibah University, Yanbu 966144, Saudi Arabia;
    tnoor@taibahu.edu.sa (T.H.N.); amars@taibahu.edu.sa (A.A.); satlam@yahoo.com (E.-S.A.)
2   Computer Science Department, Faculty of Science, Tanta University, Tanta 31527, Egypt;
    ibrahim.gad@science.tanta.edu.eg
*   Correspondence: mahmoud_osman_s@yahoo.com
†   These authors contributed equally to this work.

**Abstract:**    During the COVID-19 epidemic, Twitter has become a vital platform for people to express their impressions and feelings towards the COVID-19 epidemic. There is an unavoidable need to examine various patterns on social media platforms in order to reduce public anxiety and misconceptions. Based on this study, various public service messages can be disseminated, and necessary steps can be taken to manage the scourge. There has already been a lot of work conducted in several languages, but little has been conducted on Arabic tweets. The primary goal of this study is to analyze Arabic tweets about COVID-19 and extract people's impressions of different locations. This analysis will provide some insights into understanding public mood variation on Twitter, which could be useful for governments to identify the effect of COVID-19 over space and make decisions based on that understanding. To achieve that, two strategies are used to analyze people's impressions from Twitter: machine learning approach and the deep learning approach. To conduct this study, we scraped Arabic tweets up with 12,000 tweets that were manually labeled and classify them as positive, neutral or negative feelings. Specialising in Saudi Arabia, the collected dataset consists of 2174 positive tweets and 2879 negative tweets. First, TF-IDF feature vectors are used for feature representation. Then, several models are implemented to  identify people's impression over time using Twitter Geo-tag information. Finally, Geographic Information Systems (GIS) are used to map the spatial distribution of people's emotions and impressions. Experimental results show that SVC outperforms other methods in terms of performance and accuracy.

**Keywords:** COVID-19; Arabic tweets; machine learning; LSTM; NLP

## 1. Introduction

A novel Coronavirus that has not been previously detected has assailed the globe. COVID-19 is a newly discovered virus that first appeared in China in Wuhan city. It triggers mild diseases, such as the common cold. Individuals diagnosed with COVID-19 have reported to experience a variety of symptoms, including mild and severe ones. The first symptoms might begin within 2–14 days after the virus exposure. Individuals with the following symptoms might be suffering from COVID-19 [1].

According to [2], COVID-19 is spread across 214 territories and countries worldwide and 2 international conveyances. World Health Organization (WHO) reported 37,989,663 cases with 28,518,505 recovered cases and 1,084,345 deaths worldwide. There are also 29,602,850 closed cases and 8,386,813 active cases. As mentioned in [3] in 11 October 2020, Ministry of Health, and the National Health Emergency Operation Centre NHEOC. A total of 339,615 proven COVID-19 cases have been reported, while 321,485 have been considered recovered COVID-19 cases. There are 8708 active cases. The polymerase chain reaction (PCR) test is conducted in COVID-19 Certified Labs by Saudi CDC.

With a rapid increase in COVID-19, public opinion in any part of the world is crucial for developing policies related to security, healthcare, as well as tailoring standard protocols. Governments might never implement their policies without public assistance. Therefore, it is vital to be aware of what communities are discussing. Moreover, it is necessary to take proper actions to learn more about the virus and the ways of warding it off. A variety of indicators must be studied by the Government and other types of public service organizations.

Recently, social media platforms have been used to gather public opinion. Multiple researches showed that learning variations in people affected by changes and factors such as financial crises, viruses, and even wars. A growing number of studies have been conducted in text mining and processing. A large number of scientific findings in social media rely on COVID-19. For instance, authors from [4] work on a range of COVID-19 related tweets, using various analyses. They have already gathered about 700 GB of COVID-19 related tweets since March 2021. The availability of multiple tools to analyze brand effectiveness is one of Twitter's main benefits over Facebook. Without spending a lot of money, brands can employ a variety of tools. A huge number of alternatives are free, while the more expensive ones come with a little charge. Twitter may appear difficult and daunting at first. However, after you have gotten used to it, it will be your best friend for life. Not only does Twitter.com make it easier to engage with your consumers in real time, but technologies like Tweet Deck and Twitter for smart phones make it much easier. Because Twitter is a public platform, everybody can see each other's tweets, providing your material the push it need. In addition, Twitter serves as a real-time search engine. Twitter's advanced search is fantastic, allowing marketers to easily look for specific topics. Twitter's uptime has improved over time, and anyone may use the appropriate keyword combination to look for certain topics. Hashtags help with this search to a large degree. Twitter and the Internet provide far more tools for supporting and promoting brand memory than Facebook. I am not dismissing Facebook's marketing possibilities; yet, as a social media site, Facebook was designed for more private conversations. It is more for friends and family to connect than for businesses to engage customers. However, at the end of the day, if a brand is able to effectively manage their activity across both platforms, they will be able to reap the benefits of both worlds.

Opinions could be classified as positive/negative or positive/negative/neutral. Opinion mining is a highly discussed topic as there is a lot of unstructured data that makes developers and researchers do their own research to receive helpful insights. A text analysis of opinions can assist in understanding how people feel about different topics and events through opinions mining [5–11]. During the COVID-19 epidemic, several methods have been proposed to understand public attitudes and behaviors in the face of the pandemic [10,12–17]. Despite the challenges in receiving results using Natural Language Processing (NLP) and deep learning techniques due to the nature of the texts [2,18] yet, this field is still evolving at an incredible rate. It is critical to know the uses and limitations of different algorithms in various scenarios.

Despite the fact that a substantial amount of research has been conducted in several languages, additional research on Arabic tweets is required. There are just a few Arabic datasets for COVID-19 data. It is necessary to evaluate various social media trends in order to eliminate public misunderstandings and fear. The main novelty of this study is to look through COVID-19-regarding Arabic tweets and see what people think about different locations. This analysis is intended to aid various government and corporate entities in gaining a better understanding of public attitude and behavior in the face of the pandemic, as well as making strategic decisions in response. This paper uses the TF-IDF method to extract features from Twitter posts. Several Machine Learning and Deep Learning approaches are then applied to analyze the Twitter posts about COVID-19 and identify the distribution of polarity over different regions based on Geo-tag information. The dataset for this study was crawled using Twitter API. It consists of 2174 positive tweets and 2879 negative tweets. Tweets percentage in the dataset for the highest six Arab

countries are shown in Figure 1. Experiments show that SVM models are better at analyzing people's opinions regarding COVID-1 than other methods.

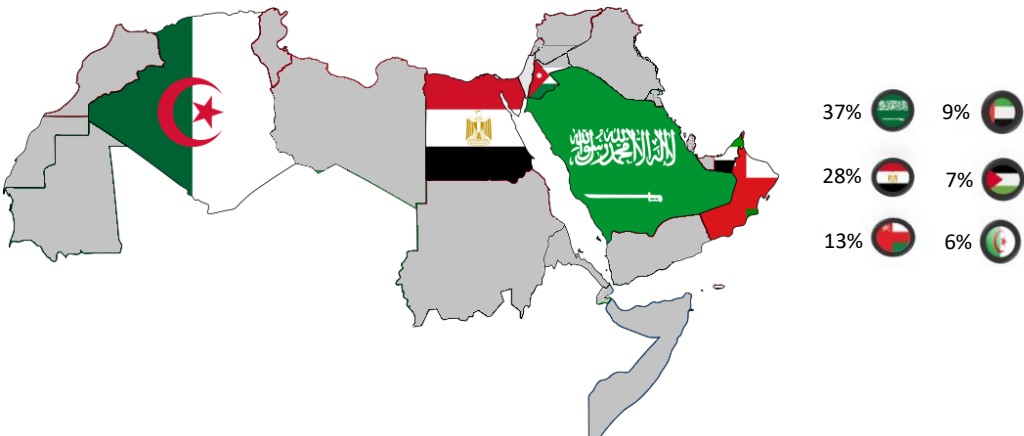

**Figure 1.** Tweets percentage in the dataset for each country.

The main points of this paper are as follows:

- We analyse Arabic Twitter posts to identify people's impressions regarding COVID-19-pandemic to aid the government in comprehending the public's perceptions and making required decisions based on them.
- The sentiment polarity patterns across different spatial zones are observed using geotag data.
- We examine various machine learning models and deep learning techniques to understand public behavior and attitudes. SVM is found to be more accurate than other methods for analyzing and monitoring COVID-19 opinions and enhanced the accuracy of the predictions.

Section 2 of this paper presents literature review. Section 3 depicts the system methodology. The data sets and system classification model as well as the performance evaluation of the suggested classification mode describes in Section 4. Finally, Section 5 is the conclusion and possible future works.

## 2. Related Work

This section includes some key contributions that help detect user behaviors and actions in diverse scenarios around the world by many researchers working on Sentiment Analysis on Twitter social media.

Kaur and Sharma [19] investigate how people feel about COVID-19 and, as a result, how different people feel about this condition. The Twitter API was used to collect Coronavirus-related tweets, which were then analyzed with machine learning algorithms to identify whether they were positive, negative, or neutral. Additionally, the Textblob dataset is utilized to analyze tweets, and the NLTK library is used to pre-process fetched tweets. Finally, several visualizations are used to present the intriguing results in positive, negative, and neutral feelings.

According to Prabhakar Kaila et al. [20], The raw data collected was appropriate and deserving of being used in COVID-19 pandemic tests. The Latent Dirichlet Allocation was applied to the data in the document term matrix from the datasets (LDA). The LDA algorithms discovered that the great majority of related information on the COVID-19 infection paramedic was negative attitudes and good sentiments like trust throughout the surgery.

Medford et al. [21] used a set of COVID-19-related hashtags to search for relevant tweets from 14 January to 28 January 2020. Tweets are extracted and saved as plain text using the API. The frequency of keywords like illness prevention strategies, immunization,

and racial prejudice is discovered and investigated. In addition, each tweet's emotional valence (positive and negative) and major emotion are determined using sentiment analysis (happy, sad, fear, joy, good or surprise). Finally, in order to detect and analyze related subjects in tweets over time, a machine learning algorithm is deployed.

Alhajji et al. [22] use NLTK module in Python to analyse impressions using the Nave Bayes model. Tweets with hashtags relevant to seven government-mandated public health activities were gathered and evaluated. Except for one measure, the data demonstrate that positive tweets outnumber negative tweets.

Anuratha [23] not only gathered data on tweets published in January 2020, but he also examined the tweets in terms of two major aspects: understanding the word occurrence pattern and sentiment recognition. Furthermore, they also noticed that the number of IDs tweeting about the Coronavirus has been steadily increasing. Many words, such as COVID-19, Coronavirus, and Wuhan city, were repeated. N-grams models such as unigram, bigram, and trigram repetition were built on top of the top thousand frequencies.

For better understanding the public attitude and concerns about COVID-19 vaccines, Hussain et al. [24] created and implemented an artificial intelligence-based approach to assess public opinion on social media in the United Kingdom and the United States against COVID-19 vaccines.

Previous studies focused on establishing algorithms for predicting the transmission of COVID-19 through Twitter with high accuracy. In previous research studies, the major flaw was that no study was conducted to analyze and monitor the emotional responses to COVID-19 over different locations vaccine. In addition, only a few contributions have been made to Arabic content analysis due to the complexity of the language. Therefore, this study analyzes and identifies people's opinion toward Arab twitter users' impressions classification during COVID-19 pandemic and helps decision makers in the future. Moreover, in this paper, several models are implemented to bidentify people's impression over time using Twitter Geo-tag information. Geographic Information Systems (GIS) are used to map the spatial distribution of people's emotions and impressions.

## 3. Methodology

Researchers have recently focused a lot of emphasis on understanding user emotions in Twitter. The purpose of this research is to see how Arabic Twitter affects people's emotions and feelings about the COVID-19 Pandemic. To properly analyze the Arabic Twitter affects, emotion analysis can be utilized to determine conveyed emotions from unstructured writings. We use Geo tag data to investigate people's emotions using a variety of machine learning and deep learning approaches. The framework architecture of the proposed model is discussed in Figure 2. There are three key stages in the proposed model: (1) pre-processing, (2) learning, and (3) emotion classification. In the sections that follow, we will go through those components in greater depth.

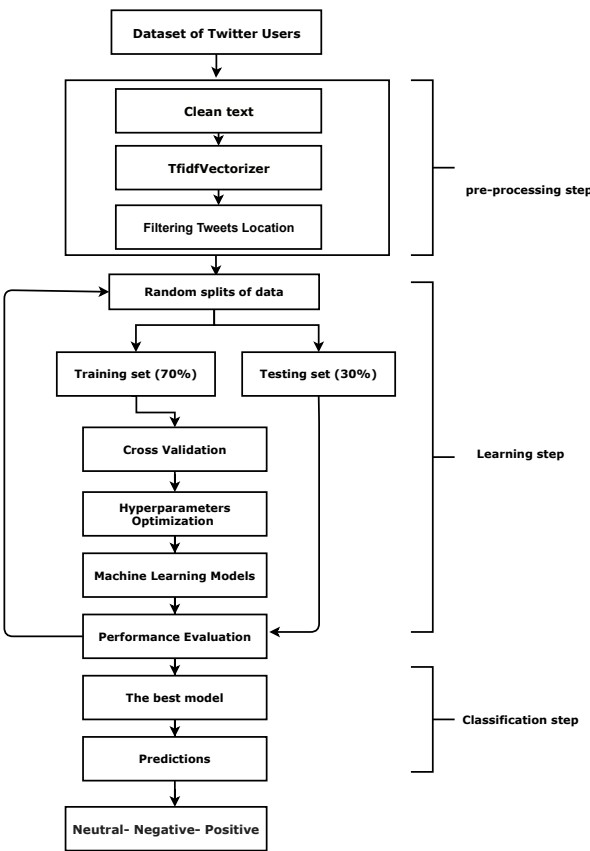

**Figure 2.** The main steps for machine learning models.

### 3.1. Pre-Processing

3.1.1. Clean Text

A crucial part of our mode is pre-processing. The tweets are cleaned as part of the pre-processing procedure to increase the quality of our model. In this paper, we collect all tweets into a Comma Separated Values (CSV) file, which is then cleaned by deleting duplicate rows or tweets that are similar. Associated and redundant symbols such as @, RT, #, URLs, numeric values, and punctuation marks are also removed from the tweets. First, words are tokenized and the most frequent terms are counted. After this, relevant features are collected from the text in the tweets to develop and train the classification model. In text data, stop-words are typically regarded as noise. There are a variety of criteria that can be used to determine whether a word is a stop-word. A word can be called a stop-word if its frequency in a document is too low. Because Arabic has a large number of lexical tokens, it has a large number of stop-words in its lexicon. Stop-words in Arabic usually contain the following characteristics [5]: As a single word, they (a) have no meaning, (b) occur frequently in sentences, and (c) serve as an influential component in completing the structure of a sentence or phrase (d) They are rarely used as keywords, and even when coupled with other stop-words, they cannot make a sentence [25]. The advantages of eliminating stop-words: It can increase algorithm accuracy since only key and meaningful words will be used for training and meaningless words will be deleted, and the model will learn more effectively as a result [26].

3.1.2. Filtering Tweets Location

Twitter provides geographical metadata for each tweet: (1) Tweet location that includes the user's location at the time of the tweet; (2) Account Location is determined by the user's home address in their public profile. Given a latitude and a longitude pair, Twitter API could return a list of tweets for specific location. Approximately 12,000 tweets are collected

in our experiment and classified as positive or negative. The number of tweets collected for each region is shown in Figure 1. The Geo-tag information is used to model the sentiment polarity patterns in different regions.

### 3.1.3. *TF-IDF* Vectorizer

Term frequency and inverse document frequency (*TF-IDF*) is a statistic metric used in text mining that determines how relevant a term is to a single document in a corpus of texts. *TF-IDF* assigns two scores to each word in a document: term frequency (*TF*) and inverse document frequency (*IDF*). *IDF* calculates a word's inverse document frequency, whereas *TF* calculates the number of times a term appears in a document. The *TF-IDF* weight of that sentence is computed by multiplying the two scores [27,28].

Where *TF-IDF* denotes the importance of a phrase in a given text. The higher the *TF-IDF* weight score, the more significant the term in the text. We use the *TF-IDF* measure to extract and model different terms in our dataset in the suggested model. We can compute *TF* and *IDF* of terms as in the following Equations (1) and (2):

$$TF(t) = \frac{\text{Number of times word appear in documents}}{\text{Total number of words in the documents}} \tag{1}$$

$$IDF = \log \frac{\text{Total number of documents}}{\text{Frequency of word in all documents}} \tag{2}$$

Here, the *TF-IDF* value is computed by making a product of the two statistics which means the weight of the words as in Equation (3):

$$TF\text{-}IDF(t) = TF(t) * IDF(t) \tag{3}$$

*TF-IDF* is used in this study because it provides a vector representation of tweets. The next step is to analyze the users' feelings about COVID-19 after we computed the *TF-IDF* for our dataset.

### 3.2. *Learning*

The tweets are tokenized into words and then turned into a feature vector throughout the learning phase. The converted data is divided into train and test sets, with 70% of the data going to the training set and 30% to the testing set. The k-fold is a typical type of cross-validation approach that uses several distinct samples of data to obtain the average accuracy metric of the model. To explore our suggested approach, various machine learning techtechniques such as [29], MultinomialNB [30], BernoulliNB [30], SGD [31], Decision tree [32], Random forest [33] and K-Neighbor [34] are employed to select the best in terms of results.

Linear Support Vector Classifier (SVC) study uses a linear kernel function to classify data and works good with large datasets [29]. The Linear SVC has more parameters, such as penalty normalization and loss function. The kernel method cannot be modified because linear SVC depend on the kernel technique. A linear SVC is designed to fit the input data, returning a "best fit" hyper-plane that divides or categorizes the data. After obtaining the hyperplane, the features are put into the classifier, which predicts the class to which it belongs.

Naïve Bayes is an algorithm that provides identical weight age to all features or attributes. Since one attribute can't impact another one, it makes the algorithm more efficient. Naïve Bayes classifier (NBC) is an easy, efficient, and proven text classification method of [30]. NBC theoretically relies on the Bayes theorem and has been applied in document classification since the 1950s. Naïve Bayes classifier employs a posterior estimation to determine the class. For instance, features are classified in accordance with the highest conditional possibility. In general, there are two models of the classifier, namely Multi-nominal Naïve Bayes (for example, binary feature representation) and Bernoulli Naïve Bayes (for example, features are shown with frequency). Multinomial Naive Bayes

Classifier (MultinomialNB) is a common machine learning application for categorical data analysis, specifically text data. It is a probabilistic learning strategy that's largely utilized in NLP (NLP). Bernoulli Nave Bayes is a statistical method that generates outputs on a boolean basis by utilizing the existence of the required text. Bernoulli Distribution, which is discrete, is fed into this classifier. When an unwanted term needs to be discovered or a specific sort of word needs to be labeled in a text, this form of Naive Bayes classifier can help. It also distinguishes itself from the multinomial technique by producing binary output in the form of 1–0, True–False, or Yes–No.

The term'stochastic' refers to a system or process that has a random probability associated with it. As a result, in Stochastic Gradient Descent (SGD), instead of selecting the entire data set for each iteration, a few samples are chosen at random. As a result, instead of finding the total of the gradients of all the examples, we get the gradient of a single example's cost function at each iteration. In short, SGD is an iterative method for finding the best smoothness qualities for an objective function (e.g., differentiable or sub-differentiable). Because it replaces the actual gradient (derived from the complete data set) with an estimate, it can be considered a stochastic approximation of gradient descent optimization (calculated from a randomly selected subset of the data). This lowers processing costs by allowing for faster iterations in exchange for a reduced convergence rate, which is especially useful in high-dimensional optimization problems [35].

Decision Tree is a type of preferred Machine Learning such that the data is separated with respect to parameters constantly. The two entities nodes and leaves can be used to explore the tree. Decisions or consequences are represented by leaves. Whereas, the data is divided at decision nodes [32]. A random forest is a collection of (many) decision trees which work together to solve a problem (ensemble learning). As a final outcome of the RF's prediction, the class with the highest votes from the trees will be selected. These decision tree models are uncorrelated in order to shield each other from their own faults. Overfitting difficulties are common with decision tree algorithms. This can be overcome with a random forest technique. It can solve regression and classification issues, and it can handle a huge number of characteristics while estimating which ones are essential in the underlying data. Without carefully prepared data modifications, random forest is capable of learning as shown in Figure 3.

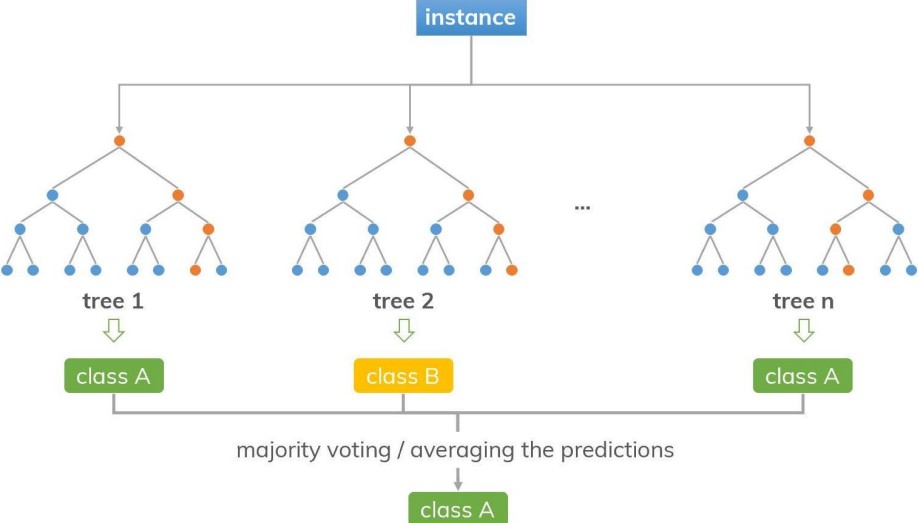

**Figure 3.** Random Forest Model with three classes [33].

K-nearest (K-NN) is an algorithm, which collects all available processes and then classified existing ones by using the distance functions (similarity metric). Since the 1970s, K-NN has been employed as non-parametric process in statistical estimation and pattern recognition. By using the majority vote of its neighbors and a distance function, the case is classified with the allocated case to the class, which has most members among

K-closest neighbors as a measuring. The outcome is affected by whether k-NN is used for classification or regression. Here, the outcome of a k-NN classification represents a class membership. the object is classified using the majority of its neighbors according to the allocated object class among its k-closest neighbors (e.g., k is to a small positive integer). Thus, the outcome of a k-NN regression is considered as the object's property value. The number refers to the average of the values of k-closest neighbors. After the function have been evaluated and only approximated locally, all computations are postponed. Since this algorithm is depending on the distance for classification, the training data is normalized to greatly enhance the accuracy [34]. If the features perform separate physical units or hook up wildly in various scales, the training data can be normalized dramatically to enhance its accuracy.Uni-gram, bi-gram, tri-gram, and word embedding using word2vec were used to evaluate the performance of the selected classifiers. The classification is carried out after the texts have been transformed into vectors.

Recurrent Neural Networks (RNNs) is the group of artificial neural networks with connections among nodes creating a directed graph. In other words, RNNs are a sequence of neural network blocks, which resemble a chain with blocks being linked to one another. Every block is sending a message to a receiver. This structure lets RNNs find out temporal behavior and get sequential information. This makes it look like a 'natural' approach to handling textual information since the text is sequential by nature as in Equation (4).

$$h(t) = f_c(h(t-1), x(t)) \tag{4}$$

where $h(t)$ refers to new states, $f_c$ is a function with respect to parameter $c$, such that $h(t-1)$ is to old state and $x(t)$ represents an input vector at $t$ (time step).

While simple RNNs can form long sequential information, theoretically they are unable to depict long sequences in real time applications [36,37]. A variety of RNNs versions are available yet we will focus on the Long Short-Term Memory (LSTM) networks as displayed in Figure 4.

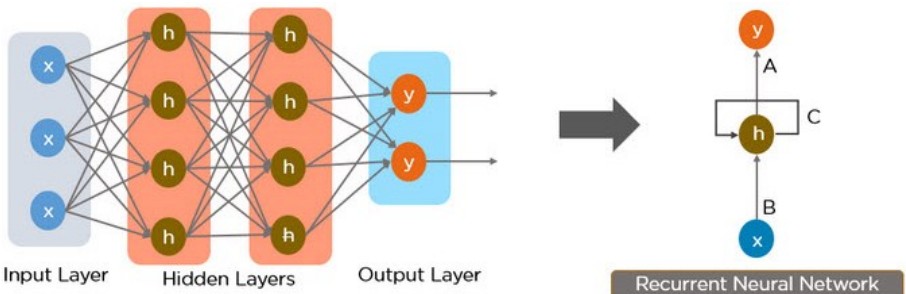

**Figure 4.** Recurrent Neural Network with respect to LSTM [36].

The relationship between distinct words in these sequences is stored in the LSTM. A lot of scientists use LSTM for various kinds of text categorization tasks [38]. LSTM is also used for COVID-19-related sentiment categorization. They received tweets data and used Latent Dirichlet Allocation (LDA) for the theme modeling as shown in Figure 5. The sigmoid function determines the LSTM Model as in Equation (5).

$$f_t = \sigma(W_f * [h_{t-1}, x_t] + b_f) \tag{5}$$

such that $f_t$ refers to the forget gate, which decides the deleted information (i.e., not important from previous time step). $h_{t-1}$ is to the previous state along with current input $x_t$. For more information, please refer to [38].

Scientists divided the sentiments into groups such as negative, very negative, positive, very positive and neutral.

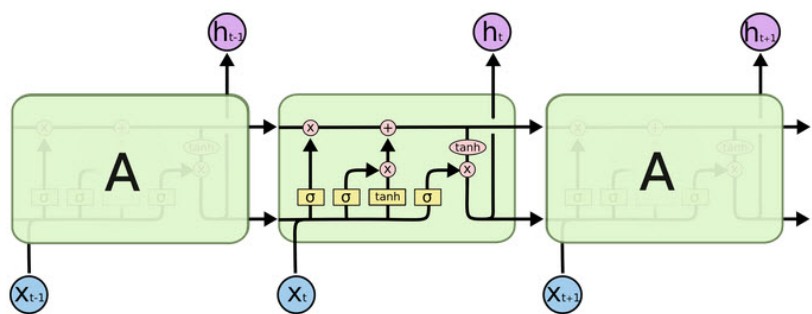

**Figure 5.** LSTM Model contains four interacting layers.

Long Short Term Memory Networks—commonly referred as LSTMs are the RNN's type, which can learn long-term dependencies [38,39]. Figure 5 diagram is taken from an interesting blog: Understanding LSTM Networks posted on 27 August 2015: https: //colah.github.io/posts/2015-08-Understanding-LSTMs/ (accessed on 23 January 2022). The problem of long-term dependency is avoided explicitly in the LSTM model. All recurrent neural networks consist of a series of iterated neural network processes. The Iterating process will have a relatively simple structure (i.e., a single tanh layer) in ordinary RNNs. Features will be passed to LSTM for training. Our data in training and testing is hldivided as 70% for training and 30% for testing. New data can be entered by users for further classification. In our work we employ three models; Naive Bayes, Long Short-Term Memory and Recurrent Neural Networks to identify which is the most effective in terms of results. The k-fold cross-validation approach is a typical kind of cross-validation that is used to obtain the average accuracy measure of a model by producing many distinct samples of data.

*3.3. Classification*

The suggested approach is based on machine learning models, which have excellent generalization and a very accurate paradigm. In our study we will explore more neural network approaches, and select the most effective one in terms of results empirically. The main rationale for utilizing ML is to solve the over-fitting problem that happens in neural networks. Furthermore, it has a risk-minimization structural principle. ML have a structure that allows them to run dichotomic classes, especially in higher-dimensional space, as well as hypothesize a maximal separation hyper-plane.

**4. Experimental Evaluation**

The goal of the model is to categorize emotions as positive, neutral, or negative. In addition, we use Geo tag information to study the impact of COVID-19 on different locations. The tweets are tokenized into words and then turned into a feature vector while the model is being trained. Long Short-Term Memory (LSTM), Naive Bayes, decision tree, and SVM are the major models built and evaluated for the framework. Natural language Tooklit (NLTK) has lately been popular due to its ability to implement a variety of standard and sophisticated machine learning algorithm. As a result, in our experiment, we compare the results of our model using the above-mentioned accessible machine leaning in NLTK.

In the subsections that follow, we first go over the dataset that was used in this work, and then provide the experimental results in both simulated and tabular form. The results suggest that our model can successfully capture people's emotions conveyed in social content by comparing the model's output with well-known methodologies.

*4.1. Data Collection*

Using web scraping and crawling is not illegal to use to scrape old tweets from Twitter. It violates the Twitter Policy. As a result, utilizing the Twitter Developer Application Programming Interface, a dataset was compiled (API). Unfortunately, because to a Twitter policy restriction, we were only able to collect about 11,000 tweets about COVID-19 that

were written in Arabic. In this step the connection to Twitter API is established to retrieve tweets related to certain keywords as Table 1. The dataset was collected from November 2020 until January 2021. The labelling process is time consuming, and to make sure the labelling process is accurate, we have hired two PhD students who are experts in this field and in emotions analysis. We have given them a clear guideline to differentiate between positive, negative or neutral. The conflicts are further addressed by the third participant, who is also an expert in this field. The final dataset after labelling process consists of 1600 tweets labelled as positive, 6000 labelled as negative, and 6000 labelled as as neutral. We have divided the dataset into 3000, 4000, 5000 and 6000 tweets. We found the performance of the proposed model is independent of the size of the data set and the performance of the model was stable.

**Table 1.** Tweets related to certain keywords.

| Word in Arabic | Word in English |
|---|---|
| كورونا | Corona |
| التباعد الاجتماعي | Social distancing |
| كوفيد ١٩ | COVID-19 |
| منظمة الصحة العالمية | World Health Organization |
| لبس الكمامات | Wear masks |
| تعقيم اليدين | Hand sanitizer |
| وباء عالمي | A global epidemic |

COVID-19 and Corona were two Arabic trending hashtags that were used to compile the tweets. The Twitter API will be utilized to acquire 30,000 tweets. Using a Python application, we turned the collected tweets into an excel file. Here, 20,000 tweets have hlbeen separated into 40 files and then distributed to various individuals to avoid repetition. A Python software will be used to collect labeled tweets, which will then be transformed into data frames. The data histogram plotted below demonstrates that the majority of Arabs had negative feelings towards the COVID-19 pandemic. According to the Figure 6, it is being noted that neutral tweets are less than 6000, bad tweets are more than 6000 and positive tweets are fewer than 1500.

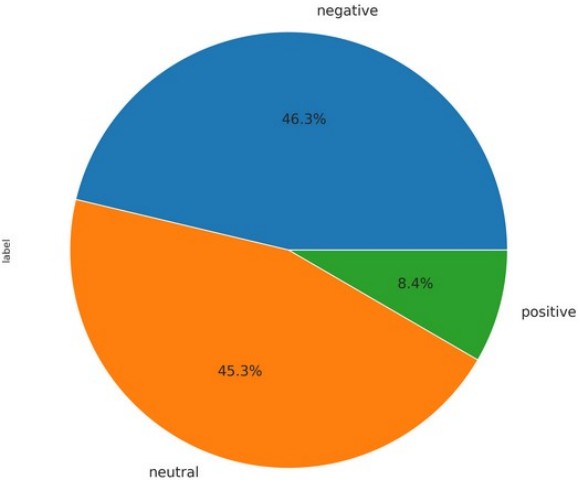

**Figure 6.** The percentage of samples for each class.

On 2 March 2020, Saudi Arabia declared the first case of COVID-19. The preliminary analysis of tweets from February 2020 reveals that there were 2879 more negative tweets (57%) than positive tweets (43%) prior to the finding of any case in Saudi Arabia.

Table 2 shows sentiments for spatial zones for tweets in many Arab countries. Several Machine Learning and Deep Learning approaches are then applied to analyze the Twitter posts about COVID-19 and identify the distribution of polarity over different regions based on Geo-tag information. From Table 2, Twitter sentiments are location specific and they might depend on the activities conducted at certain locations such as Saudi Arabia and Egypt which has the highest positive/negative/neutral opinions regarding COVD-19. Finally, Geographic Information Systems (GIS) are used to map the spatial distribution of people's emotions and impressions.

**Table 2.** Sentiments for spatial zones.

| Country (Arabic Twitter) | Positive | Neutral | Negative | Total | Evaluation (≈%) |
|---|---|---|---|---|---|
| Saudi Arabia | 325 | 1822 | 1903 | 4050 | 37% |
| Egypt | 248 | 1398 | 1459 | 3105 | 28% |
| Jordan | 119 | 668 | 698 | 1485 | 13% |
| UAE | 76 | 425 | 444 | 945 | 9% |
| Palestine | 65 | 365 | 380 | 810 | 7% |
| Algeria | 54 | 304 | 317 | 675 | 6% |
| Total | 887 | 4982 | 5201 | 11,070 | — |
| Evaluation (≈%) | 8 | 45 | 47 | — | (100%) |

Figure 7 shows the tweets percentage in the dataset by country, in which 37% of tweets were from Saudi Arabia followed by 28% from Egypt, 13% from Amman, 9% from UAE, 7% from Palestine, 6% from Algeria.

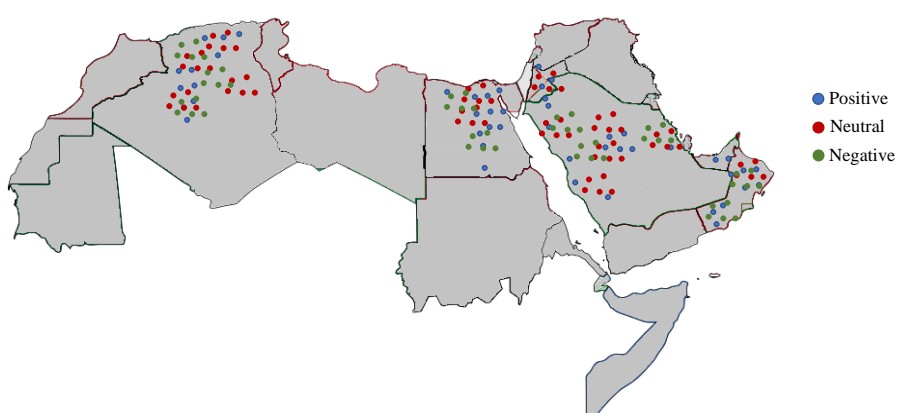

**Figure 7.** Tweets percentage in the dataset for each country.

*4.2. Performance Evaluation*

We decided to employ other performance metrics to address the issue because classification accuracy is not the optimal statistic in the case of class imbalanced datasets. The confusion matrix that used to depict the application of classifier is a table, in which the displayed values in diagonal either positive or negative are correctly classified. Here, the off-diagonal cells correspond to classified observations incorrectly. As a result, four additional performance metrics, such as accuracy, sensitivity (recall), *F1-score*, and ROC curve, are employed as extra performance assessments.

The mathematical expressions of accuracy, *F1-score*, recall, and precision determined on the basis of the False Positive (*FP*), True Positive (*TP*), False Negative (*FN*), and True Negative (*TN*) test data samples are Equations (6)–(9) [40].

$$Accuracy = \frac{TP + TN}{TP + FP + FN + TN} \tag{6}$$

$$F1\text{-}score = 2\frac{precision \times recall}{precision + recall} \tag{7}$$

$$Recall = Specificity = \frac{TP}{TP + FP} \tag{8}$$

$$Precision = \frac{TP}{TP + FN} \tag{9}$$

### 4.3. Machine Learning Models

Experiments are conducted utilizing a dataset generated using hashtags like social distancing, Corona, and COIVD-19. Around 11,000 tweets are included in the dataset. To fit the model, the converted data is separated into train and test sets, with 70% of the data going to the learning set and 30% going to the testing set. Uni-gram, bi-gram, tri-gram, and word embedding using word2vec were used to evaluate the performance of the selected classifiers. The classification process begins after the texts have been transformed into vectors. Hyperparametric classifiers including K-nearest Neighbor (KNN), Nave Bayes (NB), Stochastic Gradient Descent (SGD), Support Vector Classifier (SVC) classifier, Random Forest (RF), and Decision Tree (DT) are used in this research [40–42].

For text data classification applications, linear classifiers such as SVC and linear SVC are more efficient. A strong classifier must be constructed in order to obtain high accuracy. By creating many diverse samples of data, the K-fold Cross-Validation technique calculates the average accuracy measure of the model. The k-fold approach is a typical kind of cross-validation in which, for example, if $k = 10$, 9 folds are used for learning and 1 fold is used for testing the model, and this process is repeated until all folds have a chance to be the test set one by one. By using k-fold cross-validation, the average of the values computed in the loop is used as the performance metric. As a result, we get a strong concept and a clear knowledge of the model's generalization capabilities, which is especially beneficial when we have minimal data and cannot afford to split it into learning and testing sets.

The comparison of machine learning approaches is shown in Table 3. On the basis of different N-gram values, the comparison analysis is carried out utilizing machine learning models such as KNN, NB, SGD, DT, RF, and SVC. On the test dataset, the performance of each model is evaluated using arrangement metrics such as precision, recall, and accuracy. In terms of experimental results, the SVC model surpasses the others. The SVC model was used to divide emotions into three groups: positive, negative, and neutral. In addition, the SVC technique favors major "negative" classes. Finally, the SVC model's performance is shown in Table 3 with a maximum accuracy of 85 percent, a precision of 85 percent, and an *F1-score* of 85 percent.

Table 4 shows the values of recall, precision, and accuracy based on the classification report for SVC model. SVM models used for determining people's impressions of classes negative and neutral outperform the models used for determining positive impressions. In addition, the SVC method is biased due to its high sensitivity to major classes. The *F1-score* is also used to calculate the comprehensive measurement of performance. The SVM technique showed remarkable results in predicting Emotion from tweets, as evidenced by the performance measures values.

The confusion matrix of SVC model is given in Figure 8. The SVC model correctly predicted 3502 out of 4102 tweets, while it mis-predicted 600. The SVC classifier model had a 80 percent accuracy rate. The performance criteria of the proposed approach are given in Table 4. This model achieved 85 percent precision in the "neutral" class, 80 percent in the

"negative" class, and 76 percent in the "positive" class. In addition, the confusion matrix of MultinomialNB classifier is shown in Figure 8b. The MultinomialNB model correctly predicted 3362 out of 4102 tweets, while it mis-predicted 740.

**Table 3.** The comparison of machine learning methods.

| Classifier | N-Gram | Accuracy | Precision | Recall |
|---|---|---|---|---|
| **SVC** | 1 | 0.848 | 0.845 | 0.848 |
| | 2 | 0.854 | 0.851 | 0.854 |
| | 3 | 0.851 | 0.850 | 0.851 |
| **MultinomialNB** | 1 | 0.808 | 0.810 | 0.808 |
| | 2 | 0.820 | 0.824 | 0.820 |
| | 3 | 0.824 | 0.830 | 0.824 |
| **BernoulliNB** | 1 | 0.804 | 0.814 | 0.804 |
| | 2 | 0.822 | 0.830 | 0.822 |
| | 3 | 0.826 | 0.831 | 0.826 |
| **SGD** | 1 | 0.788 | 0.781 | 0.788 |
| | 2 | 0.810 | 0.810 | 0.810 |
| | 3 | 0.807 | 0.803 | 0.807 |
| **Decision Tree** | 1 | 0.672 | 0.730 | 0.672 |
| | 2 | 0.678 | 0.732 | 0.678 |
| | 3 | 0.656 | 0.771 | 0.656 |
| **Random Forest** | 1 | 0.738 | 0.699 | 0.738 |
| | 2 | 0.695 | 0.766 | 0.695 |
| | 3 | 0.691 | 0.769 | 0.691 |
| **KNN** | 1 | 0.797 | 0.807 | 0.797 |
| | 2 | 0.801 | 0.814 | 0.801 |
| | 3 | 0.786 | 0.795 | 0.786 |

**Table 4.** Classification report for SVC.

| | Precision | Recall | F1-Score | Support |
|---|---|---|---|---|
| **Neutral** | 0.85 | 0.91 | 0.88 | 1851 |
| **Negative** | 0.86 | 0.87 | 0.87 | 1903 |
| **Positive** | 0.76 | 0.49 | 0.59 | 348 |
| **Accuracy** | | | 0.85 | 4102 |
| **Macro avg** | 0.83 | 0.75 | 0.78 | 4102 |
| **Weighted avg** | 0.85 | 0.85 | 0.85 | 4102 |

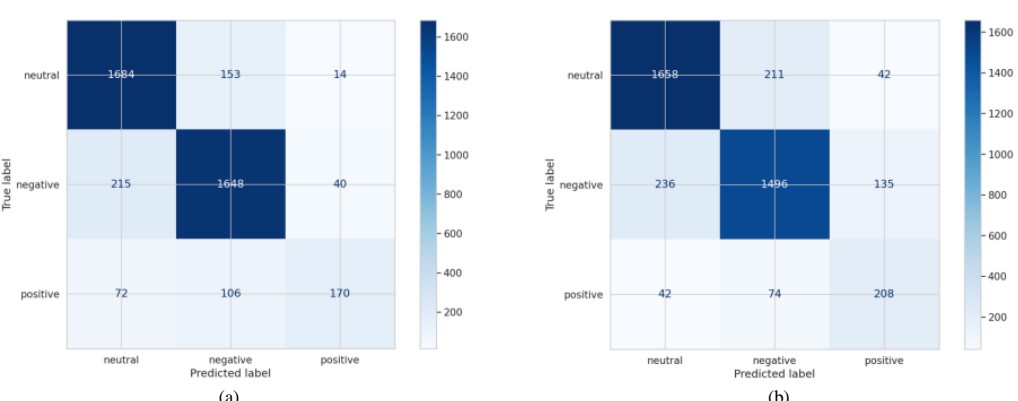

**Figure 8.** Confusion matrix of (**a**) SVC, (**b**) MultinomialNB.

*4.4. LSTM Model*

The LSTM model attempts to categorize the data into three categories: positive, neutral, and negative. The tweets are tokenized into words and then turned into a feature vector while the model is being trained. To compare the findings with advanced machine learning methods, the LSTM model is constructed and evaluated. As a result, LSTM is an excellent tool for any task that involves a sequence. Because the words that come before it establish the meaning of a word. This cleared the way for the use of neural networks in NLP and narrative analysis. The LSTM may be used to generate text. You can train the model on a writer's text, for example, and it will be able to generate new sentences that imitate the writer's style and interests. Sequence-to-Sequence For translations, LSTM models represent the current state of the art. They also have a variety of uses, such as time series forecasting. Table 5 shows the classification report for LSTM model for recall, precision and F-score. From the confusion matrix for LSTM model in Figure 9, the accuracy is the total of diagonal row divided by the total of all cells which means: accuracy = *TP + TN* / Total 707 + 361 + 25)/(707 + 48 + 5 + 483 + 361 + 9 + 106 + 32 + 25) = 0.615 approximately 62% as displayed in Table 5. As shown in Figure 9, there are three categories in the confusion matrix: negative, neutral and positive.

**Table 5.** Classification report for LSTM model.

|  | **Precision** | **Recall** | **F1-Score** | **Support** |
|---|---|---|---|---|
| **Neutral** | 0.55 | 0.93 | 0.69 | 760 |
| **Negative** | 0.82 | 0.42 | 0.56 | 853 |
| **Positive** | 0.64 | 0.15 | 0.25 | 163 |
| **Accuracy** |  |  | 0.62 | 1776 |
| **Macro avg** | 0.67 | 0.50 | 0.50 | 1776 |

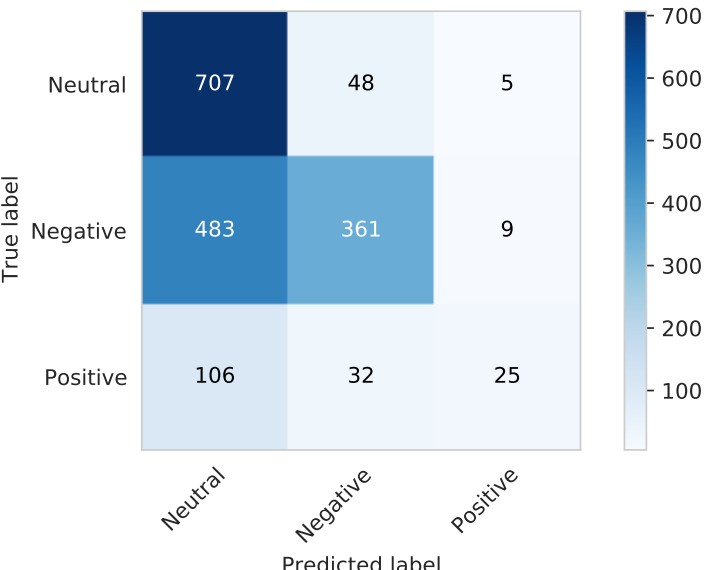

**Figure 9.** Confusion matrix for LSTM model.

The first class positive 138 is misclassified, negative 492 are misclassified, for normal 53 are misclassified. A rule of thumb the diagonal of the confusion matrix is the true values. So neutral tweets were 707 true. Negative tweets were around 361 labeled right and positive tweets were around 25 labeled right. Total tweets used for producing confusion matrix are around 1776 tweet.

1.  False Negative (*FN*): the model anticipated a negative outcome, but it was incorrect. The False-negative value for a class is the sum of the values of the linked rows, except for the *TP* value.
2.  False Positive (*FP*): the model anticipated a positive result, but it was incorrect. Except for the *TP* value, the False-positive value for a class is the total of the values in the relevant column.
3.  True Negative (*TN*): the model correctly predicted a negative outcome. The total of all columns and rows for a class, except the values of the class for which the values are being calculated, is the True Negative value for that class.
4.  True Positive (*TP*): The model correctly predicted a positive outcome. The real positive value is where the actual and expected values are the same.

The following steps are used to calculate the *TP*, *TN*, *FP*, and *FN* values for the class Neutral using Figure 9: *TP*: The real and expected values should match. The *TP* value is the value of cell 1 when it comes to neutral class. *FN*: The total of the values in the relevant rows, except the *TP* value. *FN* = (cell 2 + cell 3) = (48 + 5) = 53. *FP*: Except for the *TP* value, the sum of the values in the appropriate column. *FP* = (cell 4 + cell 7) = (483 + 106) = 589. *TN*: The sum of all columns and rows except the values of the class for which the values are being calculated. *TN* = (cell 5 + cell 6 + cell 8 + cell 9) = 361 + 9 + 32 + 25= 427. Similarly, for Negative class the values/metrics are calculated as follows: *TP*: 361 (cell 5). *FN*: 483 + 9 = 492 (cell 4 +cell 6). *FP*: 48 + 32 = 80 (cell 2 + cell 8). *TN*: 707 + 5 + 106 + 25 = 843 (cell 1 + cell 3 + cell 7 + cell 9).

The term "recall" refers to the ability to recall, i.e., How much of the positive classes did we properly predict? Recall(Neutral) = True Positive/True Positive + False Negative = 707/(707 + 48 + 5) = 0.93 approximately 93%. Similarly, for Negative and Positive classes, the recall values are 42%, and 15%. The number of positive classes that we correctly predicted are actually positive is referred to as precision. Precision (Negative) = True Positive/True Positive + False Positive = 361/(48 + 361 + 32) = 0.818 approximately its 82%. Similarly, for the Neutral and Positive classes, the recall values are 55%, and 64%. Finally, accuracy = *TP* + *TN*/Total = (707 + 361 + 25)/(707 + 48 + 5 + 483 + 361 + 9 + 106 + 32 + 25) = 0.615 approximately 62% as displayed in Table 5.

The confusion matrix of LSTM obtained is given in Figure 9. The proposed LSTM model correctly predicted 1093 out of 1776 tweets, while it mis-predicted 666. The model achieved 55% precision in the "Neutral" class, and 82% in the "Negative" class while 64% of class "Positive". The LSTM classifier model had a 62% accuracy rate. The performance criteria of the proposed approach are given in Table 5.

The limitations of the results obtained are due to the different dialects of the Arabic language. The Arabic language is a language rich in words, meanings, and different dialects. It differs from one country to another, in addition to the difference in the same country according to its various regions. Therefore, the tweets written and read differ from one place to another and from one person to another, so it is difficult to analyze them and obtain satisfactory results. There are also tweets written as a joke and using codes that people circulate according to the language spoken in that region, and therefore they have an impact on the results. In addition, the nature of tweets varies according to spatio-temporal for each country. However, this presented work is considered a basis for what is to come, and some of them are interested in this field of research, and for this we will in the future expand the database of common words and phrases in this field until we obtain promising results.

## 5. Conclusions

Since the main occurrence of COVID-19 was seen in Wuhan, China, in December 2019, this virus has sparked a global emergency. The global spread and severity of illnesses prompted the World Health Organization to declare COVID-19 a pandemic threat. Without mandatory vaccines, countries throughout the world rushed to implement a variety of preventive measures in order to restrict the spread of the infection and, as a result, avoid

a complete failure of their medical care systems. Conclusion analysis, also recognized as assessment mining, is a strong tool and an innovative technique for assessing public perceptions and commitment to important health interventions. Pandemics, such as the current coronavirus situation, are a tumultuous and rapidly evolving test that necessitates careful observation of how people perceive the impending danger and how they respond to methods and rules. In addition, assessments are necessary for the development of appropriate letter content that addresses common concerns. Finally, prior to the finding of any case in Saudi Arabia, an early study of tweets from February 2020 revealed that there were 2174 positive tweets and 2879 negative tweets. Furthermore, the analysis is conducted with an accuracy of 84% using the SVC approach. In the future, the web application can be expanded by creating mobile applications which support it. Regarding various subjects, new categories could be created. In addition, more graphs, such as the map of locating bad and positive tweets, can be added.

**Author Contributions:** Conceptualization, T.H.N. and A.A.; methodology, M.E. and I.G.; software, E.-S.A.; validation, M.E., E.-S.A. and I.G.; formal analysis, M.E.; investigation, T.H.N. and M.E.; resources, A.A.; data curation, E.-S.A.; writing—original draft preparation, T.H.N.; writing—review and editing, A.A.; visualization, M.E.; supervision, M.E. and I.G.; project administration, E.-S.A. funding acquisition, M.E. All authors have read and agreed to the published version of the manuscript.

**Funding:** This research received no external funding.

**Institutional Review Board Statement:** Not applicable.

**Informed Consent Statement:** Not applicable.

**Data Availability Statement:** All data have been present in main text.

**Conflicts of Interest:** The authors declare no conflict of interest.

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
