# Peer review of "Spatial Impressions Monitoring during COVID-19 Pandemic Using Machine Learning Techniques"

_computers, doi:10.3390/computers11040052_

Round 1
Reviewer 1 Report
General Comments
This is an interesting paper. The authors processed both machine learning and deep learning analysis for Arabic tweets about COVID-19 and extracted people’s spatial dependent impressions. The article is well organized with a smooth flow of information during the explanation of each method. It is well-written, with very few clerical errors, and the style and layout are very well articulated. Overall, the authors clearly demonstrate their approach and detail the performance gain in this research field and the article meets the required standards for publication after minor edits.
Specific Comments
- Table 2. Sentiments for spatial zones are not clear, consider adding more details to the description and legends for evaluation.
- Table 5 is not clear and has lots of empty spaces for the accuracy row.
- A related paper section should also be included, to clarify the novelty compared with previous papers.
- Why did the author choose the LSTM method, and what are the advantages of this method, it was not explained?
- Check figure 5. It is 99% similar to some published material (https://colah.github.io/posts/2015-08-Understanding-LSTMs/) Please clarify. Try to remake your figures or cite others' work properly.
- It's important to share the code of the experiments to guarantee the reproducibility of the experiments and to assure the veracity of the results.
- There were still some typos in the paper, I hope to check the full text carefully.
Author Response
Thank you for pointing out that our work could be understood in this way. Based on your valuable comments, we carried out some work on. We have got important ideas that help us towards improving the quality this work.

Reviewer 2 Report
This manuscript showed the application of machine learning and deep learning techniques to a dataset obtained from Twitter to classify texts in positive, negative or neutral. The manuscript is well-structured, authors justify the research with a good literature review, but there are some weaknesses to solve before publishing it.
Major changes:
1. Related to data collection and preprocessing, it's interesting to create new datasets and label them, but the authors must explain how they labelled them. It is very important to know the methodology to determine if a tweet is positive, negative or neutral.
2. Related to models, it's so difficult to know where is the novelty in the research. The authors said that he applied deep learning techniques, but, in my opinion, it's necessary to test how Bi-LSTM or BERT models perform with this data because they are newer models in the field of natural language processing. For example, here we can see some papers of NLP that used BERT models and show better performance than other techniques:
[1] M. P. Shahri, K. Lyon, J. Schearer, y I. Kahanda, «DeepPPPred: An Ensemble of BERT, CNN, and RNN for Classifying Co-mentions of Proteins and Phenotypes», dec. 2020. doi: 10.1101/2020.09.18.304329.
[2] F. Harrag, M. Debbah, K. Darwish, y A. Abdelali, «BERT Transformer model for Detecting Arabic GPT2 Auto-Generated Tweets», arXiv:2101.09345 [cs], Jan. 2021, Accessed: 23th January de 2022. [Online]. Available in: http://arxiv.org/abs/2101.09345
[3] Benítez-Andrades JA, Alija-Pérez JM, Vidal M, Pastor-Vargas R, García-Ordás MT Automatic Classification of Tweets about Eating Disorders: Traditional Machine Learning techniques and BERT models JMIR Medical Informatics. 01/02/2022:34492. doi: https://doi.org/10.2196/preprints.34492
The third reference appears to be in the process of publication following acceptance of the manuscript.
3. To ensure the reproducibility of the experiments, you need to share the code and data in a repository such as github or similar.
Author Response

(The authors gave the same response as above.)

Reviewer 3 Report
The paper lacks scientific contribution in the field of machine learning techniques for sentiment classification of short text expressions such as tweets. No innovative approaches are presented, nor does the comparison of the results using existing methods bring any concrete advancement in the analysis of the Arabic language.
Sentiment classification is relevant for various applications, including social, political, and media. While these examples can be used to motivate the study, they do not constitute a contribution to the field of computer science. In this sense, the Covid pandemic issue is a good running example. However, the paper claims contributions that I did not find supported by its results, e.g., the contribution of "techniques to understand public behavior" (page 2). I doubt that a computer science paper should provide means to "aid the government in comprehending the public’s perceptions and making required decisions based on them" (page 2).
The English text of the paper is in great need of improvement.
Author Response

(The authors gave the same response as above.)

Reviewer 4 Report
In this paper , two strategies are used to analyze people
impressions from Twitter: the Machine learning approach and Deep learning approach.
Why only twitter data used? I would recommend to compare the data with other social media platforms such as Facebook
The date which twitter data has been collected should be mention
In addition, the related work should be updated with more recent studies in the field such as:
Hussain, A., Tahir, A., Hussain, Z., Sheikh, Z., Gogate, M., Dashtipour, K., Ali, A. and Sheikh, A., 2021. Artificial intelligence–enabled analysis of public attitudes on facebook and twitter toward covid-19 vaccines in the united kingdom and the united states: Observational study. Journal of medical Internet research, 23(4), p.e26627.
The main novelty of the paper should be added
I recommend to add top ten bigram and trigram of the word occurances. In addition, the wordcloud should be added.
Author Response

(The authors gave the same response as above.)

Round 2
Reviewer 2 Report
I thank the authors for the changes made.
Author Response
Once again, we want to thank all the reviewers who have taken their precious time and read our paper carefully and provide us with such relevant comments that can help us in improving the quality of the research paper. We are so much grateful for you all for pointing out that our work could be understood in this way and can be considered for publication in this highly reputed Journal. At this stage, we have addressed the comments give and highlighted in the revised manuscript.
Thank you very much for your relevant comment. We have improved the quality of the manuscript with the help of English language experts.
We have add the source code in GitHub as in the following Link:
https://github.com/ibrahim85/Twitter-Users-Impressions-Classification-during-COVID-19
Reviewer 3 Report
Unfortunately, I find that the article has not been sufficiently improved for publication in Computers. No CS scientific contributions have been added to improve the merits of the article. Even the English text has not been sufficiently improved, e.g., not a single correction in the abstract.
The work presented is an exercise in the application of existing methods. It is a starting point for researching the topic. Future work should develop a novel scientific contribution by taking innovative approaches. For example, particular text processing specific to Arabic texts could be developed. As the authors consider geo-tags, they could possibly be related to dialects of certain geolocations to improve the results. Some authors' considerations relating to irony and jokes in text are not an excuse. These occur in all languages and are not specific to Arabic. This is a known challenge in semantic analysis.
What would change if the presented work analysed tweets on a different topic, e.g., war or environmental issues, rather than COVID? I do not see any particular development in the approach.
Once again, I point out that the article lacks computer science innovations. A publication in CS should present CS contributions that can then be applied by experts in various fields, including public relations and politics.
Author Response
Once again, we want to thank all the reviewers who have taken their precious time and read our paper carefully and provide us with such relevant comments that can help us in improving the quality of the research paper. We are so much grateful for you all for pointing out that our work could be understood in this way and can be considered for publication in this highly reputed Journal. At this stage, we have addressed the comments give and highlighted in the revised manuscript.
